# Chemical Composition and In Vitro Antioxidant Activity of *Sida rhombifolia* L. Volatile Organic Compounds

**DOI:** 10.3390/molecules27207067

**Published:** 2022-10-19

**Authors:** Ziyue Xu, Peizhong Gao, Dun Liu, Wenzhi Song, Lingfan Zhu, Xu Liu

**Affiliations:** 1SDU-ANU Joint Science College, Shandong University, Weihai 264209, China; 2Marine College, Shandong University, Weihai 264209, China

**Keywords:** *Sida rhombifolia* L., volatile organic compounds (VOCs), GC-MS, GC-FID, antioxidant activity

## Abstract

In the current study, the phytochemical constituents of volatile organic compounds (VOCs) obtained from *Sida rhombifolia* L. were identified by GC-FID and GC-MS analysis. A total of 73 volatile organic compounds were identified. The major components of *S. rhombifolia* VOCs were identified as palmitic acid (21.56%), phytol (7.02%), 6,10,14-trimethyl-2-pentadecanone (6.30%), oleic acid (5.48%), 2-pentyl-furan (5.23%), and linoleic acid (3.21%). The VOCs are rich in fatty acids (32.50%), olefine aldehyde (9.59%), ketone (9.41%), enol (9.02%), aldehyde (8.63%), and ketene (6.41%). The antioxidant capacity of *S. rhombifolia* VOCs was determined by 2,2-diphenyl-1-picryl-hydrazyl-hydrate (DPPH), 2,2-azinobis-(3-ethylbenzothiazolin-6-sulfonic acid) diammonium salt (ABTS), and ferric reducing/antioxidant power (FRAP) methods with butylated hydroxytoluene (BHT) and Trolox as standard. The VOCs showed dose-dependent antioxidant activity with IC50 (50% inhibitory concentration) values of 5.48 ± 0.024 and 1.47 ± 0.012 mg/mL for DPPH and ABTS assays, respectively. FRAP antioxidant capacity was 83.10 ± 1.66 mM/g. The results show that the VOCs distilled from *S. rhombifolia* have a moderate antioxidant property that can be utilized as a natural botanical supplement or an antioxidant.

## 1. Introduction

To prevent oxidation in many industries and biological systems, synthetic antioxidants are commonly used. However, studies assessing the toxicology of those synthetic compounds have demonstrated their adverse impact on human health [1]. In the face of global public health challenges owing to the overuse of chemically synthesized compounds, there has been a paradigm shift toward nature, and natural product research potentially plays a pivotal role. Faced with many stresses and environmental challenges, coupled with being sedentary, plants have developed many functional molecules to protect themselves [2]. Those plant-derived molecules may be more economical and could be exploited as effective antioxidant alternatives to offer a viable solution [3].

China has a rich flora in terms of biodiversity in Asia. Endemic and medicinal plants in this rich flora constitute a vital place. The genus Sida (Malvaceae) has about 200 species generally scattered in tropical and subtropical areas, whose pharmacological uses have been supported by several studies [4,5,6]. *S. rhombifolia* L., commonly known as teaweed and Queensland hemp, is an erect perennial shrub bearing yellowish flowers. The alternate leaves are dull green and linear–oblong to lanceolate and sometimes rhombic, with serrate margins and finely stellate hairs on the upper surface of the leaf and a dense amount of stellate hairs on the undersurface, making it look white (Figure 1). *S. rhombifolia* is a reputed Chinese herbal medicine traditionally used as a remedy for various ailments, such as stomach pain, diarrhea, gastritis, enteritis, and dysentery. In addition, stems of *S. rhombifolia* abound in mucilage and are employed as emollients and demulcents both for internal and external use. The herb is also helpful as a febrifuge with pepper [7]. In addition to ethnomedicinal observations, in vitro studies show that the hexane, acetone, and methanol extractions from *S. rhombifolia* possess anti-hyperglycemic, antioxidant, and anti-inflammatory properties [8]. Polyphenol-rich fractions from *S. rhombifolia* showed considerable antibacterial activity against cotrimoxazole-resistant bacteria strains compared with the standard antibiotic [9]. Pharmacological investigations have revealed the antinociceptive capacity of ethyl acetate extracts [10]. *S. rhombifolia* also showed potential for increased wound healing and analgesic properties [11,12,13]. Since many multifactorial diseases mentioned above are associated with oxidative stress [14], and *S. rhombifolia* is attached to numerous ethnomedicinal importance, the antioxidant activities of *S. rhombifolia* VOCs should be tested. However, comprehensive information on the chemical composition and antioxidant activities of VOCs distilled from *S. rhombifolia* is still lacking in the literature. To fill the gap, jointly with offering scientific support and a chemical basis, the present study was designed to obtain the phytochemical composition and chemically characterize the antioxidant capacity of *S. rhombifolia* VOCs.

## 2. Results

### 2.1. Volatile Components Yield and Phytochemical Characterization

The hydrodistillation of *S. rhombifolia* biomass using a Clevenger-type apparatus permitted us to obtain volatile oil with a yield of 0.2 mL from 1.5 kg of *S. rhombifolia* biomass. The chemical compositions of *S. rhombifolia* VOCs are listed in detail in Table 1 along with their retention times (RT), retention index (RI), concentration (%), and CAS ID according to their order of elution on the column in GC-MS. The ion chromatogram of *S. rhombifolia* VOCs is shown in Figure 2.

A total of 73 compounds were identified in *S. rhombifolia* VOCs, contributing 94.34% of the total volatile oils, containing a complex mixture of phytochemical components. The percentage concentrations of each compound were calculated by GC-FID data. Palmitic acid (21.56%), phytol (7.02%), 6,10,14-trimethyl-2-pentadecanone (6.30%), oleic acid (5.48%), 2-pentyl-furan (5.23%), and linoleic acid (3.21%) were found to be major components.

### 2.2. Antioxidant Properties

In the present study, the antioxidant activities of *S. rhombifolia* VOCs were evaluated by the DPPH, ABTS, and FRAP assays.

#### 2.2.1. DPPH Assay

The present study applies the test as a simple, quick, adequate test for the comparative evaluation of volatile organic compounds. The radical scavenging capacity of the *S. rhombifolia* VOCs was tested using the stable free radical DPPH^•^ [15]. Figure 3 demonstrates the effective concentrations of VOCs distilled from *S. rhombifolia* required to scavenge DPPH radical and the scavenging values (RSA%) as an inhibition percentage. The antioxidant activity increased with VOC concentrations in a dose-dependent manner. At the highest tested concentration (8 mg/mL), *S. rhombifolia* VOCs showed 62.98% antioxidant activity, while BHT and Trolox, the standard antioxidants, showed 84.70% (200 μg/μL) and 94.24% (100 μg/μL) activity, respectively.

#### 2.2.2. ABTS Assay

As seen in Figure 4, volatile organic compounds distilled from *S. rhombifolia* scavenged ABTS^•^ free radicals in a dose-dependent manner. The ABTS IC_50_ was 1.47 ± 0.012 mg/mL, as shown in Table 2.

#### 2.2.3. FRAP Assay

As shown in Table 2, the antioxidant capacity of *S. rhombifolia* VOCs evaluated by the FRAP method was 83.10 ± 1.66 mM/g, which is about 10 times lower than standard antioxidant BHT. Since ferrous ions (Fe^2+^) are considered as one of the most effective pro-oxidants in food production, the modest chelating effects of the *S. rhombifolia* VOCs would be beneficial.

## 3. Discussion

In the present study, it should be noted that the class of fatty acid (saturated fatty acid and unsaturated fatty acid) (32.50%) represents the highest percentage, followed by olefine aldehyde (9.59%), ketone (9.41%), enol (9.02%), aldehyde (8.63%), ketene (6.41%), and furan derivatives (5.93%). The classification of chemical composition is visually shown in Figure 5.

The predominance of palmitic acid signifies that the oil was of palmitic acid chemotype, which has been reported to be biochemically important. To be specific, at given concentrations, palmitic acid was proved to show selective cytotoxicity to human leukemic cells but no cytotoxicity to normal HDF (human dermal fibroblast) cells. Furthermore, human leukemic cell line apoptosis can also be induced by palmitic acid [16]. In addition, previous studies indicated that palmitic acid showed strong antimetastatic activity by restraining tumor metastasis regulation proteins and antiproliferative activity by inducing G1 phase arrest in human prostate cancer cells [17]. Previous findings indicated that palmitic acid but not palmitoleic acid could impair insulin-induced Akt (Ser^473^) by inhibiting the activity and expression of the SERCA2 gene [18]. Moreover, palmitic acid-predominate volatile oil from *Diplazium squamigerum* was determined to possess relatively strong antioxidant capacity by ORAC assay [19]. The abundance of phytol should also be noted. This acyclic monosaturated diterpene alcohol exhibited hepatotoxicity in rats, with induction of necrosis and/or apoptosis [20], antitumor activities in tumor cell lines (PC-3, MCF-7, HL, HT-29, Hs294T, A-549, and MDA-MB-231) and in rats [21]. Moreover, phytol possessed antimicrobial [22], anti-inflammatory [23], antidiabetic [24], and hypolipidemic [25] properties. The essential oil which contains a relatively high amount 6,10,14-Trimethyl-2-pentadecanone seemed to be endowed with anti-inflammatory, cytotoxic, and antibacterial activities [26,27,28]. 2-Pentylfuran has been reported in other studies to be the major component in some other plants such as *Cirsium setidens* and *Albizia lebbeck* and those 2-pentylfuran-domainated volatile oils exhibited anti-nociceptive, anti-inflammatory, antioxidant, and anti-cancer activities [29,30]. The α-β unsaturated compounds were also identified at noteworthy levels (12.43%). It was well established that α-β-unsaturated moiety-bearing compounds could activate the NRF2/KEAP1 signaling pathway, which can be described as the chief regulator of endogenous antioxidant responses to oxidative stress. Moreover, the reactivity of α-β-unsaturated moiety-bearing compounds explained the VOCs’ significant free radical scavenging activity [31]. Thus, the traditional uses of *S. rhombifolia* to treat various ailments may be due to the main components mentioned above.

Owing to the high number of species, *Sida* is a taxonomically complex genus. Hence the comparison of phytochemical components between *S. rhombifolia* and other species is needed. Much of the chemical studies were carried out focusing on polar components of *Sida* genus such as *S. acuta*, *S. spinosa*, and *S. cordifolia*. However, concerning essential oil or VOC investigations of the genus *Sida*, only one study has been published on the species *S. cordifolia*, demonstrating good capacity of anti-microorganisms including *Staphylococcus aureus*, *Candida guilliermondii*, and so forth [32].

Various assays for evaluating antioxidant activity depend on different free radical generators acting through distinct mechanisms [33]. To evaluate the effectiveness of antioxidants, several analytical methods and various substrates should be used so that more aspects of antioxidant effectiveness are covered. DPPH and ABTS radicals are relatively stable free radicals and can be reduced by sulfur-containing biomolecules such as cysteine, ascorbate, GSH, and butylated hydroxyanisole. Accordingly, ABTS and DPPH are widely used to analyze various extracts in in vitro antioxidant activity [34].

VOCs can possess relatively good antioxidants when activity is similar to or higher than reference antioxidants. However, in the present study, *S. rhombifolia* VOCs exhibited a relatively weak antioxidant effect against the DPPH free radical, as the activity of standard antioxidants BHT and Trolox at 100 μg/μL were higher than the VOCs at the highest tested concentration. The lower IC_50_ value indicates a stronger capacity of the VOCs to act as DPPH scavengers while the higher IC_50_ value indicates a relatively lower scavenging capacity. The IC_50_ of *S. rhombifolia* VOCs was 5.48 ± 0.024 mg/mL determined by the DPPH method, as shown in Table 2. The low antioxidant capacity characterized by DPPH is due to the fact that the DPPH method was used to evaluate the antioxidant capacity of phenolic compounds [35], while no phenolic, tannins, and flavonoid compounds were identified from *S. rhombifolia* VOCs.

The difference between the results of the DPPH assay and ABTS assay may be ascribed to the slightly different mechanism: Although both ABTS assay and DPPH assay are tests based on “Mixed mode” (hydrogen atom transfer, electron transfer, and proton-coupled electron transfer mechanisms may play roles in varied proportions) as described in a previous review [36], the DPPH reaction equation can be formulated primarily with respect to the HAT mechanism as mentioned above, whereas the ABTS^•^ radical is mainly based on the electron transfer mechanism [37].

The FRAP test is a typical single electron transfer method measuring the reduction from the ferric ion (Fe^3+^) complex to the ferrous ion (Fe^2+^) complex as a measure of total antioxidant activity. In addition, Cao and Prior observed no correlation between FRAP and ABTS methods [38]. Hence, the evaluation of overall antioxidant activity via multiple assays to generate the “antioxidant profile” is highly recommended by Prior et al. [39].

## 4. Material and Methods

### 4.1. Plant Material

The aerial parts of *S. rhombifolia* were collected in Jiuzhou Town, Lingshan County, Qinzhou City, Guangxi Province, China. The collections were carried out in May 2022. The species were duly identified by Prof. Hong Zhao, Marine College, Shandong University. A voucher specimen was deposited in Marine College with the following registration numbers: VS2144. The plant material of *S. rhombifolia* was packaged correctly and maintained under refrigeration (−18 °C) until VOC extraction.

### 4.2. VOC Extraction

The fresh leaves and stems (3 kg) of *S. rhombifolia* were smashed into powders and submitted to a 5 L round bottom flask. The VOCs of the plant material were extracted by hydrodistillation in a Clevenger-type apparatus for approximately 4 h [40]. To enhance the extraction yield, the VOCs were separated from the aqueous layer by diethyl ether, with an ensuing drying process via the Termovap Sample Concentrator and anhydrous sodium sulfate. The obtained VOCs joined the former into glass flasks and were stored at a low temperature (−4 °C) for further analysis.

### 4.3. Gas Chromatography–Mass Spectrometry (GC–MS) Analysis

GC–MS analysis was carried out on an Agilent gas chromatograph–mass spectrometer (7890-5975C) equipped with a fused silica capillary column, type HP-5MS (30 m × 0.25 mm × 0.25 μm). The chromatography conditions were as follows: injector temperature: 270 °C, carrier gas: helium at a flow rate of 1.10 mL/min, temperature-rising program: initial oven temperature was 60 °C, increasing by 7 °C/min to 220 °C and held stable for 6 min, then growing by 10 °C/min to 280 °C and held steady for 6 min. Then, the mass spectrometer conditions were as follows: EI: 70 ev, 230 °C, the mass scan range of 40–450 Da, and acquisition frequency of 2. Quadrupole temperature was 150 °C and 0.5 μL samples were injected.

The VOCs and n-alkane (C_8_–C_30_) were analyzed under the same conditions. The data processing was carried out by the Agilent MassHunter Qualitative Analysis 10.0 program, and the relative abundance of each compound in the VOCs was determined by peak area normalization. The identification of the VOCs’ components was carried out by calculating their retention indices (RI) in a temperature-dependent programmed condition and compared with the recorded peaks and RI existing in the spectral library (NIST/EPA/NIH 2020). The mass spectrums of target compounds were also used. The retention indexes (RI) of the identified compounds were determined by Kovat’s method [41].

### 4.4. Antioxidant Activity Determination

#### 4.4.1. DPPH Method

The radical scavenging ability against the 2,2-diphenyl-1-picryl-hydrazyl-hydrate (DPPH) was assayed. The experimental procedure was adapted from Nenadis et al. (2002) and Munteanu et al. (2021) [36,42] with some modifications. Briefly, BHT (butylated hydroxytoluene) and Trolox (6-hydroxy-2,5,7,8-tetramethylchroman-2-carboxylic acid) were used as the positive control. An amount of 100 μL of ethanol and 150 μL prepared 0.17 mmol/L DPPH were added to the microplate as control. Aliquots of 50 μL BHT solutions or VOCs at different concentrations were added to 200 μL ethanol without DPPH in 96-well microplates serving as the sample blank. Aliquots of 50 μL BHT prepared above or VOC solutions were pipetted to prepare 100 μL ethanolic DPPH in the 96-well microplate. The absorbance was measured at 516 nm by an Epoch microplate absorbance spectrophotometer after incubation of compounds to be tested for 30 min under dark conditions. The readings for each sample were recorded using the software Microplate Manager. Tests were carried out in triplicate. Radical scavenging activity (RSA%) was calculated according to the equation:$$\mathrm{RSA}\%=\left(1-\frac{A_{\mathrm{Sample}}-A_{\mathrm{Sample}\;\mathrm{Blank}}}{A_{\mathrm{Control}}}\right)\times 100\%$$
where A_Sample_ is the absorbance of the tested sample at different concentrations, A_Control_ is the absorbance of the control (ethanolic DPPH solution), and A_Sample Blank_ is the absorbance of the ethanolic sample without DPPH. IC50 was then calculated.

#### 4.4.2. ABTS^•+^ Scavenging Activity

The ABTS^•+^-scavenging capacity was measured [43] as described with minor modifications. The ABTS^•+^ free radical was produced by mixing ABTS (2,2-azinobis-(3-ethylbenzothiazolin-6-sulfonic acid) diammonium salt) in a concentration of 7.4 mmol/L with potassium persulfate (K_2_S_2_O_8_) in a concentration of 2.6 mmol/L. The mixture was stored in a dark environment at room temperature for 12 h, permitting the entire generation of free radicals. The resulting free radical solution was diluted in absolute ethanol. To determine the free radical scavenging activity, 150 μL aliquot of diluted ABTS^•+^ solution was mixed with 100 μL of sample in gradient-diluted ethanolic solutions (0.1 mg/mL, 0.25 mg/mL, 0.5 mg/mL, 1 mg/mL, 2 mg/mL, and 4 mg/mL) in a 96-well plate. Each test of a given concentration was carried out in triplicate. After 10 min of incubation, the absorbance at 734 nm was read by an Epoch microplate absorbance spectrophotometer. The percentage inhibition (inhibition%) of the tested VOCs was calculated as follows:$$\mathrm{Inhibition}\%=\frac{A_{0}-A}{A_{0}}\times 100\%$$
where A_0_ is the absorbance of 100 μL diluted ABTS^•+^ solution mixed with 150 μL ethanol at 734 nm, while A is the absorbance of 100 μL diluted ABTS^•+^ solution mixed with 150 μL sample solution at 734 nm. IC50 was then calculated.

#### 4.4.3. FRAP (Ferric Reducing/Antioxidant Power) Assay

The capacity of the VOCs of *S. rhombifolia* to reduce chelated ferric iron (Fe^3+^) was determined according to the method shown in previous studies [44,45]. A standard solution of the Trolox represents positive control. The working agent was prepared as follows, A: pH 3.6 acetate buffer solution, B: 10 mmol/L TPTZ solution, and C: 20 mmol/L Fe^3+^ solution, and the working agent was mixed at a proportion of 10:1:1, respectively. Solutions of 1 M HCl and 40 mM HCl were used to acidify the working agent. An amount of 50 μL of different dilutions of VOCs (4000 μg/mL, 2000 μg/mL, 1000 μg/mL, 500 μg/mL, 250 μg/mL, and 100 μg/mL) and 0.25 mg/mL Trolox solution (2 μL, 5 μL, 10 μL, 15 μL, and 20 μL) were mixed with 200 μL FRAP working reagent in a 96-well microplate. The blank solution was prepared similarly by replacing VOCs with distilled water. All tests were performed in triplicates and averaged. After 30 min of reaction time, the absorbance of the resulting solution was measured at 593 nm by an Epoch microplate absorbance spectrophotometer. The concentration of Fe^2+^-TPTZ (antioxidant capacity) was calculated by comparing the absorbance at 593 nm with the standard curve of the Trolox standard solutions.

## 5. Conclusions

The present study investigated the chemical composition of the volatile oil obtained from *S. rhombifolia* growing in China. The major components of *S. rhombifolia* VOCs were identified as palmitic acid (21.56%), phytol (7.02%), 6,10,14-trimethyl-2-pentadecanone (6.30%), oleic acid (5.48%), 2-pentyl-furan (5.23%), and linoleic acid (3.21%). The chemical profile of the VOCs was dominated by fatty acids, unsaturated aldehydes, and ketones. The VOCs showed dose-dependent antioxidant activity with IC50 values of 5.48 ± 0.024 mg/mL and 1.47 ± 0.012 mg/mL for DPPH and ABTS assays, respectively. FRAP antioxidant capacity was 83.10 ± 1.66 mM/g. The VOCs of *S. rhombifolia* exhibited moderate antioxidant activity by ABTS, and FRAP methods and low antioxidant capacity by the DPPH method.

Future research involving VOCs of *Sida* species may employ the reported outcomes of this study as valuable chemotaxonomic markers to unravel the infrageneric evolutionary correlations across this particularly unique genus. Nevertheless, the outcomes of in vitro antioxidant activity reported in this study demand verification with in vivo assays before considering using these volatile organic compounds in human care.

## Figures and Tables

**Figure 1 molecules-27-07067-f001:**
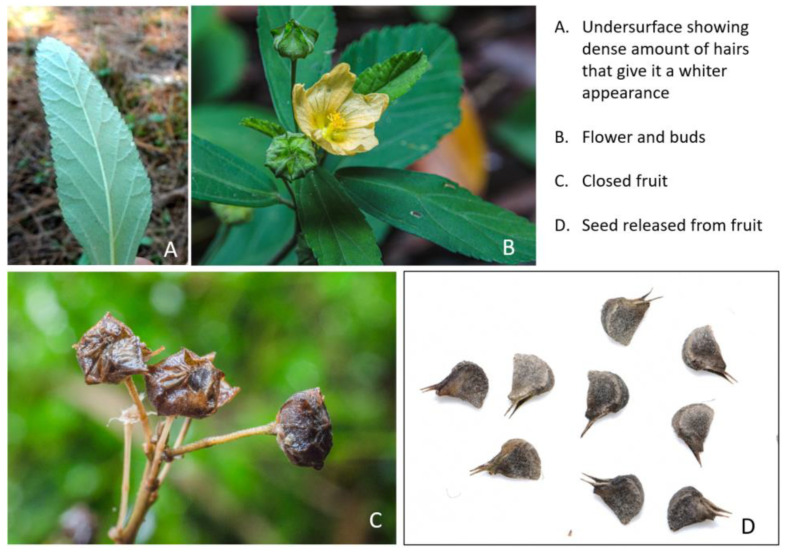
The morphological photographs of tested *S. rhombifolia* showing the leaves, flowers, fruits, and seeds released from fruits.

**Figure 2 molecules-27-07067-f002:**
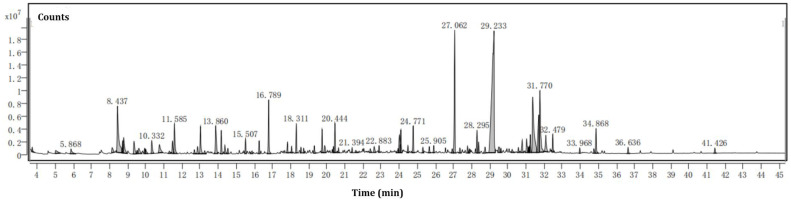
Ion chromatogram of *S. rhombifolia* volatile organic compounds derived from GC–MS.

**Figure 3 molecules-27-07067-f003:**
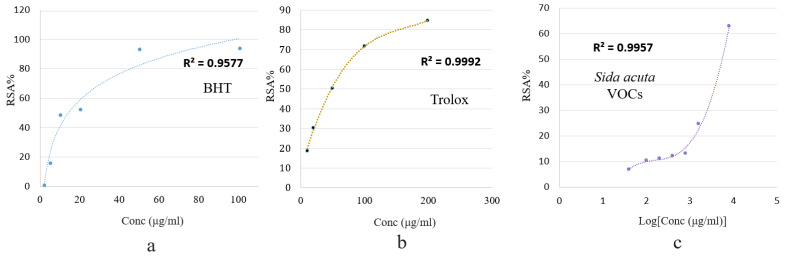
Percentage of radical scavenging activity (RSA%) of BHT (**a**), Trolox (**b**), and *S. rhombifolia* volatile organic compounds (**c**), by the DPPH method.

**Figure 4 molecules-27-07067-f004:**
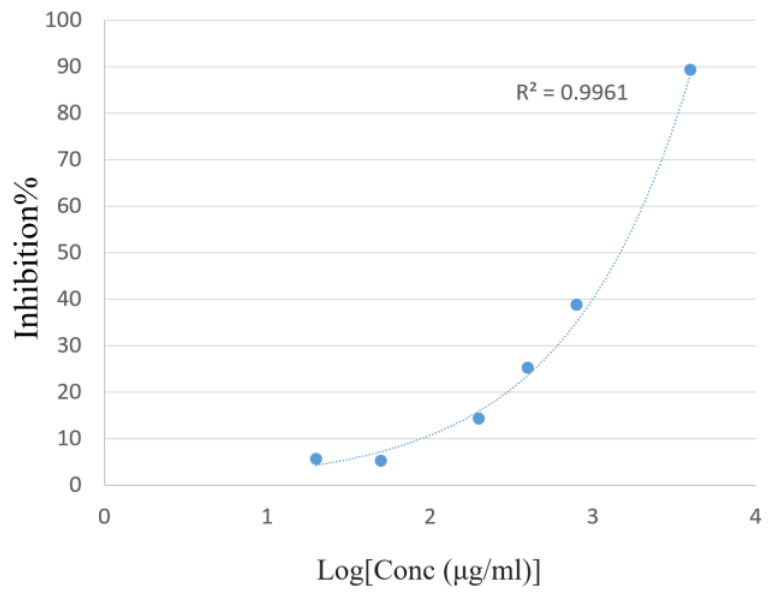
Percentage of the inhibitory effect of *S. rhombifolia* volatile organic compounds in ABTS free radical scavenging assay.

**Figure 5 molecules-27-07067-f005:**
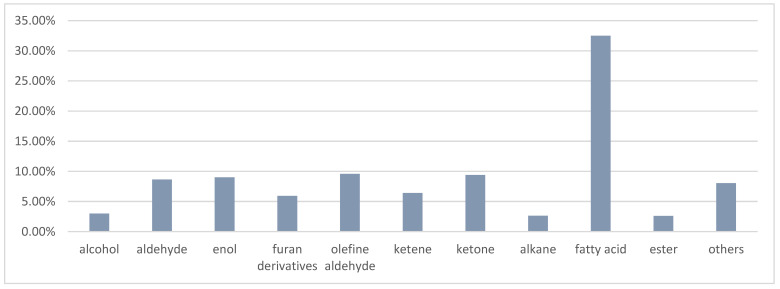
The percentage composition of chemical composition classes distilled from *S. rhombifolia*.

**Table 1 molecules-27-07067-t001:** Chemical composition of VOCs distilled from *S. rhombifolia*.

No.	RT	RI ^a^	RI ^b^	Compound	Area (%)	Identification Method	CAS ID
1	5.022	878	874	1-Hexanol	0.54%	RRI, MS	111-27-3
2	5.868	907	907	Heptanal	0.55%	RRI, MS	111-71-7
3	8.143	987	981	1-Octen-3-ol	0.53%	RRI, MS	3391-86-4
4	8.437	995	990	2-Pentyl-furan	5.23%	RRI, MS	3777-69-3
5	8.721	1005	1003	trans-2-(2-Pentenyl) furan	0.70%	RRI, MS	70424-14-5
6	8.781	1007	1004	Octanal	1.54%	RRI, MS	124-13-0
7	9.348	1029	-	2-Decyne	1.12%	MS	2384-70-5
8	9.512	1036	1022	Eucalyptol	0.26%	RRI, MS	470-82-6
9	9.615	1039	1031	2-Ethylhexanol	0.71%	RRI, MS	104-76-7
10	9.932	1051	1038	Benzeneacetaldehyde	0.33%	RRI, MS	122-78-1
11	10.332	1065	1055	2-Octenal, (E)-	0.95%	RRI, MS	2548-87-0
12	10.75	1079	1172	1-Nonanol	1.02%	RRI, MS	143-08-8
13	11.47	1103	1103	Linalool	0.83%	RRI, MS	78-70-6
14	11.585	1108	1112	Nonanal	1.99%	RRI, MS	124-19-6
15	12.681	1152	1143	5-Ethyl-6-methyl-3-hepten-2-one	0.28%	RRI, MS	57283-79-1
16	12.856	1159	1153	(E,Z)-2,6-Nonadienal	0.52%	RRI, MS	557-48-2
17	13.02	1165	1157	(E)-2-Nonenal	1.71%	RRI, MS	18829-56-6
18	13.249	1174	1161	endo-Borneol	0.26%	RRI, MS	507-70-0
19	13.86	1196	1193	2-Decanone	2.34%	RRI, MS	693-54-9
20	14.165	1208	1205	Decanal	1.18%	RRI, MS	112-31-2
21	14.367	1217	1218	(E,E)-2,4-Nonadienal	0.57%	RRI, MS	5910-87-2
22	14.531	1225	1223	β-Cyclocitral	0.27%	RRI, MS	432-25-7
23	15.507	1266	1267	(Z)-2-Decenal	0.72%	RRI, MS	2497-25-8
24	16.26	1296	1295	(E,Z)-2,4-Decadienal	0.64%	RRI, MS	25152-83-4
25	16.789	1320	1326	(E,E)-2,4-Decadienal	2.83%	RRI, MS	25152-84-5
26	17.826	1368	1366	2-Undecenal	0.54%	RRI, MS	2463-77-6
27	18.055	1378	1376	Farnesane	0.31%	RRI, MS	3891-98-3
28	18.311	1389	1384	β-Damascenone	1.32%	RRI, MS	23726-93-4
29	18.568	1400	1400	Tetradecane	0.24%	RRI, MS	629-59-4
30	18.715	1407	1404	6,10-dimethyl-2-undecanone	0.26%	RRI, MS	1604-34-8
31	19.31	1436	1436	β-Copaene	0.42%	RRI, MS	18252-44-3
32	19.735	1457	1453	6,10-dimethyl-5,9-undecadien-2-one	1.07%	RRI, MS	689-67-8
33	19.877	1464	1463	4,11-Dimethyltetradecane	0.54%	RRI, MS	55045-12-0
34	20.297	1483	1483	octahydro-4a,7,7-trimethyl-, cis-2(1*H*)-Naphthalenone	0.14%	RRI, MS	7056-56-6
35	20.368	1487	1488	Curcumene	0.31%	RRI, MS	644-30-4
36	20.444	1490	1491	trans-β-Ionone	1.43%	RRI, MS	79-77-6
37	21.977	1568	1571	trans-Nerolidol	0.19%	RRI, MS	40716-66-3
38	22.043	1571	1576	3,7,11-trimethyl-1-dodecanol	0.18%	RRI, MS	6750-34-1
39	22.397	1589	1589	(E,E)-Pseudoionone	0.25%	RRI, MS	3548-78-5
40	22.616	1599	1600	Hexadecane	0.45%	RRI, MS	544-76-3
41	22.883	1614	1614	Tetradecanal	0.34%	RRI, MS	124-25-4
42	23.936	1671	1667	6,9-Heptadecadiene	0.34%	RRI, MS	81265-03-4
43	24.012	1675	1678	Bulnesol	1.36%	RRI, MS	22451-73-6
44	24.089	1679	1680	13-Methyltetradecanal	1.40%	RRI, MS	75853-51-9
45	24.476	1699	1700	2-Pentadecanone	0.38%	RRI, MS	2345-28-0
46	24.771	1716	1715	Pentadecanal	1.36%	RRI, MS	2765-11-9
47	25.66	1767	1769	Myristic acid	0.36%	RRI, MS	544-63-8
48	26.549	1818	1815	Hexadecanal	0.21%	RRI, MS	629-80-1
49	26.925	1840	1840	Neophytadiene	0.25%	RRI, MS	504-96-1
50	27.062	1848	1846	6,10,14-Trimethyl-2-pentadecanone	6.30%	RRI, MS	502-69-2
51	27.345	1865	1877	Pentadecanoic acid	0.32%	RRI, MS	1002-84-2
52	27.771	1890	1878	(E)-2-Hexadecenal	0.40%	RRI, MS	22644-96-8
53	27.924	1899	1900	1,2-Epoxyoctadecane	0.17%	RRI, MS	7390-81-0
54	28.295	1922	1916	Farnesylacetone	1.23%	RRI, MS	1117-52-8
55	28.382	1928	1926	Hexadecanoic acid, methyl ester	0.54%	RRI, MS	112-39-0
56	28.737	1950	1947	Isophytol	0.45%	RRI, MS	505-32-8
57	29.233	1980	1975	n-Hexadecanoic acid	21.56%	RRI, MS	57-10-3
58	29.489	1996	1994	Hexadecanoic acid, ethyl ester	0.55%	RRI, MS	628-97-7
59	29.609	2003	2010	(Z)-9-Octadecanal	0.28%	RRI, MS	2423-10-1
60	30.231	2044	2042	Oxacyclooctadecan-2-one	0.27%	RRI, MS	5637-97-8
61	30.575	2067	2069	Heptadecanoic acid	0.28%	RRI, MS	506-12-7
62	30.788	2080	2075	Linoleyl methyl ketone	0.83%	RRI, MS	29204-24-8
63	31.033	2096	2093	Methyl linoleate	1.01%	RRI, MS	112-63-0
64	31.131	2102	2098	Methyl linolenate	0.28%	RRI, MS	301-00-8
65	31.192	2106	2106	γ-Palmitolactone	0.29%	RRI, MS	730-46-1
66	31.38	2119	2113	Phytol	7.02%	RRI, MS	150-86-7
67	31.704	2141	2131	Linoleic acid	3.21%	RRI, MS	60-33-3
68	31.77	2146	2152	Oleic acid	5.48%	RRI, MS	112-80-1
69	32.092	2168	2177	Octadecanoic acid	1.28%	RRI, MS	57-11-4
70	33.968	2298	2300	Tricosane	0.25%	RRI, MS	638-67-5
71	34.748	2356	2364	4,8,12,16-Tetramethylheptadecan-4-olide	0.27%	RRI, MS	96168-15-9
72	36.636	2498	2500	Pentacosane	0.31%	RRI, MS	629-99-2
73	41.426	2898	2900	Nonacosane	0.29%	RRI, MS	630-03-5

Concentration calculated from total ion chromatogram. RI ^a^: calculated retention index. RI ^b^: retention index obtained from mass spectral database. RRI: relative retention indices calculated against n-alkanes. Identification method based on the relative retention indices (RRI) of authentic compounds on the HP-5MS column. MS, identified based on computer matching of the mass spectra with Nist/EPA/NIH 2020 Mass Spectral Database and comparison with literature data.

**Table 2 molecules-27-07067-t002:** Antioxidant activities of *S. rhombifolia* for VOCs expressed as IC50 values (mg/mL) for DPPH, ABTS, and FRAP assays.

Samples	DPPH 50% Effective Concentration (mg/mL)	ABTS 50% Effective Concentration (mg/mL)	FRAP Antioxidant Capacity (mM/g)
*S. rhombifolia* VOCs	5.48 ± 0.024	1.47 ± 0.012	83.10 ± 1.66
BHT	0.042 ± 0.002	0.006 ± 0.001	
Trolox	0.015 ± 0.001	0.014 ± 0.001	

## Data Availability

The data presented in this study are available on request from the corresponding author.

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
