# Peer review of "Chemical Composition and In Vitro Antioxidant Activity of Sida rhombifolia L. Volatile Organic Compounds"

_molecules, 2022, doi:10.3390/molecules27207067_

Round 1

Reviewer 1 Report

The submitted paper 'Chemical Composition and in vitro Antioxidant Activity of Sida rhombifolia L. Volatile Organic Compounds' is interesting and could be published in Molecules, however, some improvements are required.

The paper should be prepared according to Instructions for Authors: Introduction, Results, Discussion, Material and Methods, Conclusions, References

line 98 - concentration of DPPH should be added

line 132 - should be pH

line 115 - subscripts should be included

line 220 - concentration 8 mg/µl - is it correct i.e. 8 g/ml?

line 135-136 is it a volume of Trolox (Trolox is solid),

Author Response

Dear Editor and Reviewer 1,

Thank you for the review of our manuscript "Chemical Composition and in vitro Antioxidant Activity of Sida rhombifolia L. Volatile Organic Compounds". Your constructive comments are appreciated very much. We have responded to the comments point by point and made revisions accordingly.

(0) The paper should be prepared according to Instructions for Authors: Introduction, Results, Discussion, Material and Methods, Conclusions, References

Response:Thanks a lot and modification was finished. Due to the order adjustment of each part (From Introduction, Material and Methods, Results, Discussion, Conclusions, References to Introduction, Results, Discussion, Material and Methods, Conclusions, References), the line numbers you noted were changed.

(1) line 98 - concentration of DPPH should be added

Response: Thanks a lot for the comments. From our experimental data records, the concentration of DPPH solution is 0.17 mmol/L. Corresponding data highlighted in red was added in our new manuscript.

(2) line 132 - should be pH and line 115 - subscripts should be included

Response: Thanks for your careful inspection and we have corrected those mistake.

(3) line 220 - concentration 8 mg/µl - is it correct i.e. 8 g/ml?

Response: Thanks for your comment. From our experimental method guidelines and data records, the EO was dissolved in ethanol and the resulting concentration was 20 mg/ml. Then, 100 μL mother solution (20 mg/ml) was added in microwell plate, with ensuing addition of 150 DPPH ethanolic solution. A total of 250 μL solution was mixed. As a result, the max tested EO concentration was 20 mg/ml ÷ 2.5 = 8 mg/ml, instead of 8 mg/µl or 8 g/ml.

(4) line 135-136 is it a volume of Trolox (Trolox is solid)

Response: Concentration of Trolox solution is 0.25mg/mL. Thanks for your reminder!

Hopefully, these responses and the revised manuscript can fulfil the requirements and expectations of the Editor and the Reviewers. The revised manuscript is submitted for consideration for publication. Thanks again for all of the comments and suggestions!

Sincerely,

Ziyue Xu

On behalf of all co-authors of the paper

Reviewer 2 Report

Comments and Suggestions for Authors

Manuscript ID: molecules-1941192

Article Title: Chemical Composition and in vitro Antioxidant Activity of Sida rhombifolia L. Volatile Organic Compounds

Comments:

In this manuscript authors described the chemical composition of Sida rhombifolia volatile orpganic compounds by using GC-FID and GC-MS analysis. The authors reported the identification of 78 phytocompounds and the dominant compounds were: palmitic acid (21.56%), phytol (7.02%), 6,10,14-trimethyl-2-pentadecanone (6.30%), oleic acid (5.48%), 2-pentyl-furan (5.23%), linoleic acid (3.21%). The VOCs are rich in fatty acids (32.50%), olefine aldehyde (9.59%), ketone (9.41%), enol (9.02%), aldehyde (8.89%), ketene (6.41%). Authors studied also the antioxidant activity of the obtained volatile oil by using DPPH, ABTS, and FRAP methods as compared to BHA and Trolox as standard molecules. The obtained results showed that the obtained VOCs showed dose-dependent antioxidant activity with IC50 values of 5.48 ± 0.024 and 1.47 ± 0.012 mg/mL for DPPH and ABTS assays, respectively. FRAP antioxidant capacity was 83.10 ± 1.66 mM/g.

The study is well designed, the methods used to validate the results are adequate and accurate. The introduction can be improved, and some additional  references are needed. The results are clearly presented with good tables with statistical analyses and figures with good resolution (Can be improved). The conclusion section is supported by the validation of the two techniques propped to authenticate saffron from Iran in comparison to Spain origin (especially).                     

However, there are some modifications required to be done before it is accepted for publication. The following are the specific comments on the manuscript:

Major concern:

The major concern of this manuscript is the discussion part: even f it the first time that Sida rhombifolia L. essential oil is described from this plant species, authors should discuss their results with those related to Sida genus (including species cordifolia, tuberculate, actua, spinosa ….). Authors should also add some data related the yield of extraction. Authors should add some data related the phenol, tannins, and flavonoids content of the tested essential oil.

Authors should discuss the correlation between the obtained compounds and the in vitro antioxidant activity of the tested essential oil?

Specific comments:

  1. Please explain the abbreviation used in the abstract: IC50, DPPH, FRAP, BHA …
  2. Please add a botanical description of the plant species tested with relevant picture.
  3. Please add some numbers in the Abstract and Conclusion parts to make them more scientifically sound
  4. Many paragraphs are two long and need additional references.
  5. Sida rhombifolia L.: have to be in italic in the whole manuscript, please use S. rhombifolia after the first citation.

Author Response

Dear Editor and Reviewer 2,

Thank you for the review of our manuscript "Chemical Composition and in vitro Antioxidant Activity of Sida rhombifolia L. Volatile Organic Compounds". Your constructive comments are appreciated very much. We have responded to the comments point by point and made revisions accordingly.

(1) Authors should discuss their results with those related to Sida genus.

Response: Thanks a lot for the comments. By bibliographic retrieval, only one study was published on Sida cordifolia EO antibacterial activity studies and no studies on EO chemical composition, which could be because the EO yield is relatively low. However, a few studies were carried out focusing on polar compounds extracted by solvent instead of hydrodistillation method. Hence, from our perspective, comparing polar compounds extracted from other taxonomically related species with small polar or non-polar EO compounds from Sida rhombifolia is not scientifically necessary. So, in our modified manuscript, only one EO research of Sida genus was added in citation.

(2) Authors should also add some data related the yield of extraction

Response: Thanks a lot for the comments. Extraction yield of Sida rhombifolia EO was shown in result part : “The hydrodistillation of Sida rhombifolia biomass using a Clevenger-type apparatus permitted us to obtain volatile oil with a yield of 0.2 ml from 1.5 kg of Sida rhombifolia biomass.”

(3) Authors should add some data related the phenol, tannins, and flavonoids content of the tested essential oil.

Response: Thanks a lot for the comments. However, no phenol, tannins, and flavonoids compounds was identified from our EO sample, which could partially explain the low antioxidant activity as you noted “discuss the correlation between the obtained compounds and the in vitro antioxidant activity of the tested essential oil”.

(4) Please explain the abbreviation used in the abstract: IC50, DPPH, FRAP, BHA.

Response: Thanks a lot for the comments. Lots of antioxidants were used in We use positive standard BHT instead of BHA in manuscripts and our experiments. We have explain those abbr in our modified abstract.

(5) Please add a botanical description of the plant species tested with relevant picture.

Response: The morphological photographs and description of tested species Sida rhombifolia was added in manuscripts.

(6) Many paragraphs are too long and need additional references.

Response: Thanks a lot for the comments. We have discussed the “too long” paragraphs. The manuscript has a set of logical steps. We are afraid of that cutting manuscript rashly would have side effects on the quality of the article to be sent to [molecules]. If some irrelevantly paragraphs or sentences could be noted by reviewer, we would appreciate your constructive advice and reduce the length of corresponding paragraph.

(7) Sida rhombifolia L.: have to be in italic in the whole manuscript, please use S. rhombifolia after the first citation.

Response: Thanks a lot for the comments. We have modified those mentioned above in our new manucript.

Hopefully, this response letter and the revised manuscript can fulfil the requirements and expectations of the Editor and the Reviewers. The revised manuscript is submitted for consideration for publication. Thanks again for all of the comments and suggestions!

Sincerely,

Ziyue Xu

On behalf of all co-authors of the paper

Round 2

Reviewer 2 Report

Dear authors 

Special thanks for your reply

Good luck

Author Response

Thanks for your constructive comments again!